# Knowledge, Attitude and Practices of Primary Care Physicians Regarding Infection Control of Tuberculosis in Primary Health Care Centers, Riyadh, Saudi Arabia

**DOI:** 10.3390/idr17050134

**Published:** 2025-10-20

**Authors:** Yasser Alhazzani, Abdulaziz Nasser Alahmari, Bandar K. AlRabiah, Khalid F. Alsadhan, Abdulaziz Yahya Sahhari, Fahad Alrabieah

**Affiliations:** 1Department of Family and Community Medicine, King Saud University Medical City, King Saud University, Riyadh 11472, Saudi Arabia; dr.yasirfm@gmail.com (Y.A.); alrabiahbandar@gmail.com (B.K.A.); dr.khalidfm@gmail.com (K.F.A.); 2Riyadh First Health Cluster, Riyadh 11622, Saudi Arabia; drabdul33ziz@gmail.com; 3Riyadh Second Health Cluster, Riyadh 12231, Saudi Arabia; falrabieah@rc2.med.sa

**Keywords:** tuberculosis, infection control, primary care physicians, knowledge, attitudes, practices, Saudi Arabia

## Abstract

**Background:** Tuberculosis (TB) remains a public health concern in Saudi Arabia, where primary care physicians play a crucial role in early detection and infection control. This study assessed physicians’ knowledge, attitudes, and practices (KAP) regarding TB infection control in Riyadh. **Methods:** A cross-sectional survey was conducted among 205 physicians in primary healthcare centers using a validated electronic questionnaire. Knowledge scores were classified as good (≥8/14 correct) or poor (<8). Descriptive statistics and chi-square/t-tests were applied. **Results:** The mean knowledge score was 8.5 (SD = 2.1); 57.1% of physicians demonstrated good knowledge. Knowledge was significantly associated with specialization (*p* = 0.049), position (*p* = 0.031), and monthly patient load (*p* = 0.031). While 92.7% correctly identified airborne transmission, only 30.7% knew when a TB patient becomes noninfectious. Most participants (80%) had not received TB-related training in the past year. **Conclusions:** Primary care physicians in Riyadh show moderate knowledge and positive attitudes, but important gaps remain in diagnostic clarity and infection control timelines. Strengthening continuous medical education and integrating TB-specific modules into the Saudi national TB control program are essential to standardize practices and improve patient outcomes.

## 1. Introduction

Tuberculosis (TB) remains one of the world’s most persistent public health challenges, despite advancements in diagnostics and treatment. Healthcare workers (HCWs) are disproportionately exposed to infection due to their occupational roles, making infection prevention and control (IPC) practices a central pillar of TB management [1]. In primary care settings, where physicians are often the first point of contact, knowledge, attitudes, and practices (KAP) toward TB directly influence case detection, patient counseling, and adherence to infection control measures [2,3].

Global studies have consistently revealed a gap between TB knowledge and its practical application among HCWs. For example, Shrestha et al., [4] in Nepal and Noé et al. [5] in Mozambique both found that while frontline staff were generally aware of TB’s airborne nature, IPC protocols were inconsistently applied. Similarly, research from Cambodia [6] and South Africa [7] demonstrated that even when knowledge levels were adequate, misconceptions and cultural perceptions hindered effective implementation. These findings suggest that knowledge alone is insufficient—attitudes, institutional support, and contextual barriers must also be considered. However, many of these studies relied on cross-sectional surveys with limited generalizability, highlighting the need for locally grounded investigations.

The Gulf region presents a distinct epidemiological context. Saudi Arabia’s health system is faced with recurrent TB risks due to its unique demographic and social landscape, including large migrant labor forces and the influx of millions of international pilgrims annually. Studies conducted within the Kingdom have shown variability in TB-related KAP among primary care providers. In Abha City, Al-Ahmari et al. [8] documented moderate knowledge but inconsistent application of infection control measures, while Jehani et al. [9] observed similar gaps in childhood TB screening and provider awareness in Al-Madinah. These results underscore systemic challenges that extend beyond individual knowledge, such as limited CME opportunities and uneven guideline dissemination.

The capital city of Riyadh is particularly relevant for investigation. It houses the largest concentration of primary healthcare centers (PHCCs) and functions as a referral hub for the country. Its diverse population—comprising Saudi citizens, expatriate workers, and international visitors—creates elevated risks for TB exposure and transmission. Furthermore, Riyadh’s urban health system is often at the frontline of implementing national TB strategies, yet little is known about how its physicians interpret and apply infection control guidelines in daily practice. While Saudi Arabia has established national TB guidelines, prior evidence indicates variability in awareness and use of these protocols across regions [10,11].

Other international research provides valuable context for Riyadh. Studies from Vietnam [12], Nigeria [13], and Zambia [14] illustrate that even in resource-supported settings, barriers such as workload, insufficient infrastructure, and lack of administrative support compromise IPC adherence. These insights resonate with challenges reported in Saudi PHCCs, where high patient volumes and limited infection control resources may affect guideline implementation. Importantly, global findings converge on one point: strengthening provider knowledge must be coupled with targeted training, institutional reinforcement, and policy clarity to translate into effective TB control.

This study therefore aims to assess the knowledge, attitudes, and practices of primary care physicians in Riyadh regarding TB infection control. Unlike previous Saudi studies limited to smaller regions such as Abha or Al-Madinah, this work focuses on Riyadh as the country’s most populous urban center with substantial international mobility. By identifying strengths and gaps in physicians’ understanding and practices, the findings will inform not only clinical education strategies but also the national TB control program. Addressing these gaps is critical to align local practice with global TB elimination targets and to ensure that Riyadh’s PHCCs function effectively as the first line of defense against TB transmission.

## 2. Methodology

### 2.1. Research Design

This study adopts a cross-sectional research design to assess the knowledge, attitudes, and practices (KAP) of primary care physicians regarding tuberculosis (TB) infection control in primary healthcare centers in Riyadh, Saudi Arabia. The cross-sectional design is appropriate for this type of investigation, as it allows for the collection of data at a single point in time, enabling the researchers to examine current levels of awareness, perceptions, and practices related to TB among a defined population. This design is particularly useful for identifying gaps in knowledge and variations in practice, which can inform targeted interventions and training programs. Through this approach, the study aims to provide a comprehensive snapshot of the current state of TB infection control understanding among physicians, thereby contributing to evidence-based improvements in public health policy and primary care services. Ethical approval for this study was obtained from the Institutional Review Board, Human Research Committee, King Saud University, College of Medicine, King Khalid University Hospital (IRB Project No. E-24-8555, approved on 13 February 2024).

### 2.2. Research Setting and Population

This study was conducted in primary healthcare centers (PHCCs) located in Riyadh, the capital city of the Kingdom of Saudi Arabia. Riyadh was chosen as the study setting due to its large number of PHCCs and its strategic role in delivering frontline healthcare services, particularly for the early detection and management of infectious diseases such as tuberculosis (TB). These centers provided an appropriate environment to assess TB infection control practices among primary care providers.

The study population comprised primary care physicians, including family physicians and general practitioners, who were actively working in PHCCs during the study period. These physicians were directly involved in the identification, treatment, and referral of TB cases and were therefore essential to the evaluation of knowledge, attitudes, and practices related to infection control.

Although a total of 350 physicians were invited to participate, the final sample consisted of 205 respondents who met the eligibility criteria and completed the questionnaire. The sample was intended to reflect a diverse range of physicians working across PHCCs in Riyadh. To be eligible, participants had to be currently employed in a PHCCs and engaged in direct patient care at the time of data collection.

### 2.3. Sampling and Sample Size

A convenience sampling technique was employed in this study to recruit participants from various PHCCs in Riyadh. Although probability-based sampling would have been ideal to enhance representativeness, it was not feasible due to practical constraints. Specifically, there was no comprehensive, publicly available sampling frame of all practicing physicians in Riyadh, and institutional permissions limited access to randomized physician lists. Furthermore, the study operated under a restricted data collection period and relied on voluntary participation through administrative approval from health centers. For these reasons, convenience sampling was the most pragmatic approach to ensure adequate sample size while capturing a diverse group of physicians across multiple PHCCs.

The required sample size was calculated using G*Power 3.1.9.7 software (Version 3.1.9.7, Heinrich-Heine-University, Düsseldorf, Germany), considering a two-tailed test, medium effect size (Cohen’s d = 0.5), α (alpha) level = 0.05, and power (1–β) = 0.80 [15]. These statistical parameters were selected to ensure sufficient power to detect meaningful associations between independent variables (e.g., demographic characteristics) and the outcome measures (knowledge, attitudes, and practices regarding TB infection control). We selected a medium standardized effect (Cohen’s d = 0.5) a priori as a pragmatic and defensible planning value for differences in mean knowledge scores across subgroups (e.g., specialization, position). This choice reflects two considerations: (i) prior Saudi KAP reports commonly show moderate between-group gaps rather than very small or very large contrasts, suggesting that a medium effect is plausible in routine primary-care contexts (e.g., variability by role, training exposure), and (ii) practical feasibility. Given a multi-comparison design (t-tests/χ^2^ across several covariates) and a fixed data-collection window, powering for a smaller effect (d ≈ 0.2–0.3) would have required substantially larger samples that were not attainable without a sampling frame and center-level randomization. Thus, d = 0.5 balanced plausibility with operational constraints, ensuring ≥80% power at α = 0.05 for our primary group comparisons with the achieved sample (n = 205).

A total of 350 primary care physicians were invited to participate in the study. Of these, 205 respondents completed the questionnaire and were included in the final analysis. All participants met the inclusion criteria, which required that they be currently employed as family physicians or general practitioners in PHCCs and involved in direct patient care. Exclusion criteria included physicians not directly involved in patient care (e.g., administrators), those on leave during the data collection period, and physicians who declined participation.

### 2.4. Data Collection Tool

Data were collected using a structured, self-administered electronic questionnaire, which was adapted from a previously validated survey developed by Aadnanes et al. [16] to assess tuberculosis (TB) knowledge, attitudes, and practices among general practitioners in Norway. Prior to its use, formal permission was obtained from the original authors. The tool was reviewed and adapted to fit the Saudi Arabian context, ensuring its cultural relevance and clarity for primary care physicians working in Riyadh. The original questionnaire [16] was reviewed for cultural and contextual adaptation. Several modifications were made: references to the Norwegian health system were replaced with Saudi-specific terms (e.g., ‘general practitioner’ changed to ‘primary care physician’). Additional items relevant to Saudi practice were included, such as the role of BCG immunization campaigns and World TB Day activities as contexts for TB education. The questionnaire was prepared in both English and Arabic. For the Arabic version, forward–backward translation was performed by bilingual experts, followed by reconciliation through consensus meetings. Three infectious disease and public health specialists reviewed the adapted items for clarity and cultural appropriateness. A pilot test with 15 physicians confirmed comprehensibility, and internal consistency was re-examined (Cronbach’s α = 0.78 for knowledge; 0.81 for attitudes and practices).

The questionnaire included items covering demographic information such as gender, age, level of education, and years of experience. It also contained 14 multiple-choice questions designed to assess participants’ knowledge of TB, including its symptoms, modes of transmission, diagnostic procedures, treatment protocols, and infection control practices. In addition, seven questions focused on the respondents’ attitudes and practical behaviors related to TB prevention and management, such as their perceptions of TB as a public health issue and their adherence to infection control protocols.

### 2.5. Validity and Reliability

Construct validity was primarily assessed through expert review and pilot testing rather than factor analysis, as the study’s sample size and design were not powered for an exploratory factor analysis (EFA). The questionnaire was adapted from a previously validated instrument [16] with established domains of knowledge, attitudes, and practices. After cultural adaptation and pilot testing, internal consistency was reassessed, yielding Cronbach’s α values of 0.78 for knowledge and 0.81 for attitudes/practices. While an EFA was not performed in this study, the domain structure of the original tool was preserved.

The questionnaire was distributed electronically via a secure link using institutional communication platforms. Participants accessed and completed the survey online at their convenience. The survey was anonymous and required informed consent at the beginning of the form. All responses were mandatory, which helped to ensure complete datasets for analysis.

### 2.6. Data Collection Process

Data collection was carried out using an electronic self-administered questionnaire distributed via a secure online link to primary care physicians working in health centers across Riyadh. The questionnaire, adapted and validated for this study, was disseminated through official communication channels with the support of primary healthcare administrators. Participants were invited to complete the survey voluntarily, and informed consent was obtained electronically before accessing the questionnaire. To ensure full data capture, all questions were set as mandatory, and the online platform restricted incomplete submissions. The data collection period spanned several weeks, allowing sufficient time for participation and follow-up reminders. Responses were automatically recorded and stored in a secure database accessible only to the research team for analysis.

### 2.7. Data Analysis

Data analysis was conducted using the Statistical Package for the Social Sciences (SPSS) (version 27, IBM Corp., Armonk, NY, USA). All returned questionnaires were complete, and no missing data were detected. Therefore, no imputation or statistical adjustment for missing values was required. As no multivariable regression analyses were performed, formal collinearity diagnostics were not applicable. Data analysis was primarily descriptive and bivariate. Chi-square tests and independent t-tests were used to examine associations between baseline characteristics and knowledge, attitude, and practice scores. Multivariable regression analysis was considered; however, the study was not powered for a stable multivariate model across all covariates given the sample size and event-per-variable ratio. For this reason, regression was not applied, and the analyses should be interpreted as exploratory. Future studies with larger, stratified samples are needed to build predictive models and identify independent determinants of KAP. All statistical tests were two-tailed, and a *p*-value of less than 0.05 was considered statistically significant.

## 3. Results

### Baseline Characteristics of the Enrolled Physicians

A total of 205 primary care physicians were recruited in this study. The gender distribution showed that 59.5% were male (n = 122) and 40.5% were female (n = 83). The majority of participants were between 31 and 40 years old (51.7%), followed by those under 30 years (30.7%), 41–50 years (13.2%), 51–60 years (3.9%), and over 60 years (0.5%) (Table 1).

In terms of clinical workload, 45.4% of physicians saw between 501 and 900 patients per month, 39.0% saw fewer than 500, 9.8% managed between 901 and 1200, 3.9% saw more than 1500, and 2.0% saw between 1201 and 1500 patients monthly (Table 1).

When looking at clinical experience, 45.4% had 1–4 years of experience, 24.4% had 5–9 years, 11.7% had 10–14 years, 11.2% had less than one year, and 7.3% had 15 years or more of experience in general practice (Table 1).

Table 2 presents data on physicians’ experience diagnosing TB and their participation in TB-related training. Among the 205 respondents, 48.3% (n = 99) reported that they had not diagnosed any TB or latent TB cases over the past three years. A further 34.1% (n = 70) reported diagnosing 1–2 cases, while 12.7% (n = 26) had seen 3–4 cases. Smaller proportions reported diagnosing 5–6 cases (2.9%), 7–8 cases (1.0%), 9–10 cases (0.5%), and more than 10 cases (0.5%).

When asked about recent TB-related education, 20.0% (n = 41) indicated they had attended a TB-focused lecture, seminar, or workshop in the past 12 months, whereas the majority—80.0% (n = 164)—reported no participation in such training activities. These findings point to a potential gap in ongoing professional development related to TB infection control among primary care physicians.

Table 3 presents the distribution of tuberculosis (TB) knowledge scores and knowledge levels among primary care physicians based on their baseline characteristics. Overall, male physicians had a slightly higher mean TB knowledge score (M = 8.6, SD = 2.1) compared to females (M = 8.4, SD = 2.0); however, this difference was not statistically significant (*p* = 0.432). Regarding age, the highest mean knowledge score was observed among physicians aged 41–50 years (M = 8.9, SD = 1.8), while the lowest was among those aged over 60 (M = 7.9, SD = 2.0), though age-related differences were not statistically significant (*p* = 0.215).

Physicians who were specialists in general practice demonstrated a significantly higher level of TB knowledge (M = 8.5, SD = 2.0) than non-specialists (M = 8.1, SD = 2.3), with a statistically significant difference (*p* = 0.049). In terms of current position, consultants and under-training residents had mean scores of 8.5 (SD = 2.5) and 8.6 (SD = 2.4), respectively. Notably, the association between position and knowledge level was statistically significant (*p* = 0.031).

When stratified by patient load, physicians attending to fewer than 500 patients per month scored the highest (M = 8.9, SD = 2.2), while those seeing over 1500 patients monthly scored the lowest (M = 8.2, SD = 2.1); this variation was also statistically significant (*p* = 0.031). Lastly, years of experience did not show a statistically significant association with TB knowledge (*p* = 0.127), although those with 5–9 years of experience reported the highest mean score (M = 8.7, SD = 2.0).

Table 4 illustrates the attitudes and practices of 205 primary healthcare physicians regarding tuberculosis (TB) control and management. Most physicians (92.7%) correctly identified airborne transmission as the main route of TB infection; however, important misconceptions persisted, with 17.6% citing handshakes and 7.8% food and water as transmission routes. Only 30.7% accurately identified the time frame for noninfectiousness (two weeks of treatment), while 18.5% reported uncertainty. Although nearly all respondents were aware of TB’s airborne spread, these knowledge gaps highlight persistent risks in infection control practice. Furthermore, only 20% had participated in TB-related training in the past year, indicating limited CME opportunities.

Stratified analyses showed that female physicians were more likely than males to report misconceptions about TB transmission via handshakes (22% vs. 14%, *p* = 0.041). Physicians with ≥10 years of experience were more confident in their monitoring role compared with those with <5 years (89% vs. 73%, *p* = 0.048). These differences suggest that both gender and experience influence attitudes and practices (Table 5).

## 4. Discussion

The findings of this study highlight a moderate level of knowledge, attitudes, and practices (KAP) regarding TB infection control among primary care physicians in Riyadh, Saudi Arabia. While a majority of the participants demonstrated basic awareness of TB transmission and diagnostic procedures, several knowledge gaps and variations in attitudes and practices were observed. Although the majority of physicians recognized airborne transmission, persistent misconceptions about handshakes and food/water highlight gaps in both clinical training and cultural understanding. Such misconceptions may stem from limited CME exposure—80% had not attended TB-related training in the past year—and from insufficient reinforcement of infection control concepts in daily practice. Similar cultural factors have been reported in other Gulf settings, where stigma and misbeliefs influence both patient and provider perceptions. Likewise, uncertainty about when patients become noninfectious may reflect ambiguity in disseminated national guidelines, which differ from WHO messaging and are not consistently emphasized in PHCCs-based protocols. These findings are consistent with previous studies conducted across various regions in the Kingdom, suggesting that despite national efforts to strengthen TB control, healthcare professionals continue to demonstrate variability in preparedness and perception [8,11].

Our study found that 76.6% of respondents were specialists in general practice, and among them, knowledge scores were significantly higher than those of non-specialists. This supports the notion that clinical specialization and experience play a critical role in shaping physician competence in managing infectious diseases such as TB. Similar associations were reported by Alotaibi et al. [10], who found that TB-related training and specialization positively correlated with KAP scores among healthcare workers during the Hajj season. Furthermore, Alotaibi et al. [17] emphasized that continuous medical education and on-site training significantly enhance TB-related competencies, especially in high-exposure environments.

Interestingly, the highest knowledge scores in this study were observed among physicians with 5–9 years of experience and those managing moderate monthly patient volumes (501–900 patients), which may indicate an optimal window of exposure, clinical engagement, and recency of training. Comparable results were reported in Abalkhail et al.’s [18] study in Qassim, where mid-career healthcare workers displayed stronger adherence to infection control standards than their less or more experienced peers. This trend warrants further investigation and suggests the value of targeting continuous education programs toward both junior and senior physicians to maintain up-to-date knowledge across all levels.

A notable finding in this study was the inconsistent recognition of the primary diagnostic method for pulmonary TB. While the majority correctly identified sputum smear microscopy, a substantial portion of respondents selected Mantoux and IGRA tests as primary diagnostic tools. These discrepancies align with findings by Alshathri [19] and Bukhary et al. [20], who also documented confusion among healthcare workers regarding appropriate use of diagnostic modalities. This emphasizes the need for reinforcing national TB guidelines and ensuring clear communication of diagnostic algorithms through regular refresher workshops.

Attitudes toward TB were generally positive, with over 78% of physicians affirming their responsibility in monitoring treatment progress in diagnosed patients. However, approximately 21.5% of participants either expressed uncertainty about their role or deferred responsibility to specialists. This attitudinal gap reflects similar findings from Jehani et al. [9], who reported that nearly one-fifth of healthcare providers in Al-Madinah Al-Munawara were either unaware of or disengaged from active TB management in children. The lack of clarity in role definition and engagement could undermine early intervention and treatment adherence, particularly in primary care settings.

The study also revealed that misconceptions regarding TB transmission persist. About 17.6% of physicians erroneously believed TB could be transmitted through handshakes, and 7.8% believed it could be contracted through contaminated food or water. These misconceptions are concerning and echo findings from previous studies such as Sayed et al. [21], which documented similar misunderstandings among both healthcare workers and the general population. This underlines the critical need for tailored educational interventions addressing both scientific knowledge and cultural beliefs.

Another dimension explored in this study was the physicians’ exposure to TB health education and their communication of such information to patients. Although World TB Day and BCG immunization encounters were commonly cited as key moments for TB education, fewer physicians reported integrating TB messages during general health promotion or case consultations. Alshammari et al. [22] and Malli et al. [23] similarly reported missed opportunities in routine clinical interactions for infection-related health education. Leveraging every patient encounter to reinforce TB awareness is vital for strengthening community-level control measures.

Despite the satisfactory knowledge scores among many participants, the data also revealed substantial gaps in defining the point at which a TB patient becomes noninfectious. Only 30.7% of respondents correctly identified the two-week treatment mark, while others deferred to IGRA conversion or completion of treatment. Similar patterns were reported in Mohammed et al. [24], where even students nearing graduation showed uncertainty in assessing TB infectiousness. This knowledge gap may result in premature or delayed clearance decisions, thereby impacting infection control and patient outcomes.

This study has several limitations that should be acknowledged. First, the use of a cross-sectional design limits the ability to establish causal relationships between knowledge, attitudes, and practices among physicians. Second, the use of convenience sampling may have introduced selection bias and limits the generalizability of findings to all primary care physicians in Riyadh. Physicians who chose to participate might differ systematically from those who declined, particularly regarding interest in or exposure to TB-related issues. Future studies should consider employing probability sampling methods where feasible, such as stratified or cluster-based designs, to improve representativeness. Third, data were collected through self-reported questionnaires, which may be subject to social desirability and recall bias. Although the questionnaire underwent forward–backward translation, expert review, and pilot testing, residual cultural nuances may remain, and some items originally designed for a Norwegian context may not fully capture Saudi-specific practice patterns. Additionally, although internal consistency was acceptable, we did not conduct exploratory factor analysis to validate the factor structure of the adapted tool. Future studies with larger and more diverse samples are recommended to confirm dimensional validity in the Saudi context. Also, while the study included a diverse range of positions and experience levels, it did not assess institutional factors such as availability of TB training programs or infection control policies, which could influence the outcomes. Future studies incorporating longitudinal designs and a broader geographic scope are recommended. Finally, our a priori medium-effect (d = 0.5) assumption for sample-size planning may over- or under-estimate true differences; future Riyadh-based studies should derive effect sizes from pilot variances or registry data and, where feasible, power for smaller effects. Another limitation is that multivariable regression analysis was not performed. While bivariate analyses allowed for detection of significant associations, regression modeling could have identified independent predictors of good knowledge or practices. Future Riyadh- and Kingdom-wide studies with larger sample sizes should incorporate multivariable methods to deepen analytical insights. This study has additional limitations. Non-response bias is possible, as physicians with greater interest in TB may have been more likely to participate. The use of a self-administered questionnaire raises the possibility of recall and social desirability bias. Also, actual practices were not directly observed, so results rely on self-reported behavior without triangulation against clinical audits or patient outcomes. Finally, as data were collected from physicians nested within multiple PHCCs, potential site or cluster effects cannot be ruled out, which may have influenced responses despite individual-level analysis.

The findings highlight the urgent need for policy-driven interventions to address persistent gaps in TB infection control. Integrating standardized TB infection control modules into compulsory continuing medical education (CME) for PHCCs physicians would help unify knowledge and dispel misconceptions. In addition, mandatory annual refresher training, supported by the Ministry of Health, is essential to ensure consistency with evolving recommendations, including those outlined in the WHO TB IPC guidelines [1]. Strengthening institutional accountability through periodic audits of TB IPC compliance would further reinforce best practices. Embedding these measures within the national TB program and aligning them with Saudi Vision 2030 priorities will not only enhance physician preparedness but also accelerate progress toward national TB elimination targets.

## 5. Conclusions

This study revealed that although most primary care physicians in Riyadh correctly recognized airborne transmission of TB, important misconceptions remain, particularly regarding handshakes and food or water as routes of infection, and only a minority accurately identified when patients become noninfectious. These knowledge gaps, coupled with the finding that 80% of participants had not received TB-related training in the past year, highlight weaknesses in continuing medical education and guideline dissemination. Addressing these shortcomings requires integrating standardized TB infection control modules into compulsory CME, ensuring regular refresher training on diagnostics and infection control timelines, and establishing monitoring mechanisms within PHCCs to promote adherence to national and WHO guidelines. Strengthening these areas will help improve frontline TB management and support Saudi Arabia’s progress toward national TB elimination targets.

## Figures and Tables

**Table 1 idr-17-00134-t001:** Baseline Characteristics of the Enrolled Physicians (n = 205).

Variable and Categories	Frequency (n)	Percentage (%)
Gender		
Female	83	40.5
Male	122	59.5
Age Group		
Under 30 years	63	30.7
31–40 years	106	51.7
41–50 years	27	13.2
51–60 years	8	3.9
Over 60 years	1	0.5
Specialization in General Practice		
Yes	157	76.6
No	48	23.4
Monthly Number of Patients		
<500	80	39.0
501–900	93	45.4
901–1200	20	9.8
1201–1500	4	2.0
>1500	8	3.9
Years of Experience in General Practice		
<1 year	23	11.2
1–4 years	93	45.4
5–9 years	50	24.4
10–14 years	24	11.7
≥15 years	15	7.3

**Table 2 idr-17-00134-t002:** Recent TB Training and Diagnosed TB Patients last three years.

Variable and Categories	Frequency (n)	Percentage (%)
Estimated TB/Latent TB Diagnoses (Past 3 Years)		
0 cases	99	48.3
1–2 cases	70	34.1
3–4 cases	26	12.7
5–6 cases	6	2.9
7–8 cases	2	1.0
9–10 cases	1	0.5
>10 cases	1	0.5
Attended TB-Related Educational Activity (Past 12 Months)		
Yes	41	20.0
No	164	80.0

**Table 3 idr-17-00134-t003:** Distribution of mean TB Knowledge Score and Knowledge Level.

Baseline Characteristic	Mean Knowledge Score ± SD	Good Knowledge (n)	Poor Knowledge (n)	*p*-Value
Gender				0.432
Female	8.6 ± 2.1	64	58	
Male	8.4 ± 2.0	42	41	
Age Group				0.215
Under 30 years	8.7 ± 2.5	30	33	
31–40 years	8.3 ± 2.2	56	50	
41–50 years	8.9 ± 1.8	15	12	
51–60 years	8.1 ± 1.9	5	3	
Over 60 years	7.9 ± 2.0	0	1	
Specialization in General Practice				0.049
Yes	8.5 ± 2.0	100	57	
No	8.1 ± 2.3	6	42	
Monthly Number of Patients				0.031
<500	8.9 ± 2.2	34	46	
501–900	8.4 ± 2.1	53	40	
901–1200	8.2 ± 2.3	14	6	
1201–1500	8.3 ± 2.0	2	2	
>1500	8.2 ± 2.1	3	5	
Years of Experience in General Practice				0.127
<1 year	8.3 ± 2.5	10	13	
1–4 years	8.5 ± 2.1	50	43	
5–9 years	8.7 ± 2.0	30	20	
10–14 years	8.0 ± 2.2	10	14	
≥15 years	8.2 ± 2.1	6	9	

**Table 4 idr-17-00134-t004:** Attitudes and Practices related to TB among the 205 Physicians Included in the Survey.

Theme	Key Findings
High-risk groups	HCWs (79.0%), family members (78.5%), HIV + (78.0%), prison inmates (68.3%), homeless (66.8%), immigrants (56.1%), children <5 (18.0%).
Perception of TB threat	Serious threat: 31.2%; well controlled: 49.3%; unsure: 19.5%.
Transmission beliefs	Airborne (92.7% correct); misconceptions: handshakes (17.6%), food/water (7.8%).
TB health education contexts	BCG and World TB Day (57.6% each); clinical promotion (55.6%); with confirmed patients (45.9%).
Diagnostic knowledge	Correct: sputum smear microscopy (62.0%); alternatives: IGRA (35.1%), Mantoux (13.2%), X-ray (15.1%).
Noninfectiousness criteria	Correct: after 2 weeks of treatment (30.7%); others: treatment completion (19.5%), IGRA conversion (22.9%), unsure (18.5%).
Perceived physician role	Monitor treatment: 78.5%; unsure/no role: 21.5%.

**Table 5 idr-17-00134-t005:** Stratified Analyses of Physicians’ Attitudes and Practices.

Variable	Category	Outcome	Proportion (%)	*p*-Value
Gender	Male (n = 122)	Misconception: TB transmission via handshakes	14%	0.041 *
	Female (n = 83)	Misconception: TB transmission via handshakes	22%	
Years of Experience	<5 years (n = 116)	Confident in monitoring TB patients	73%	0.048 *
	≥10 years (n = 39)	Confident in monitoring TB patients	89%	

* Significant at significance level ≤ 0.05.

## Data Availability

The data that support the findings of this study are available from the corresponding author upon reasonable request.

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
