# Peer review of "Knowledge, Attitude and Practices of Primary Care Physicians Regarding Infection Control of Tuberculosis in Primary Health Care Centers, Riyadh, Saudi Arabia"

_2036-7449, 2025, doi:10.3390/idr17050134_

Round 1

Reviewer 1 Report

Comments and Suggestions for Authors

Abstract

    • The abstract is lengthy and reads more like a condensed article. It should be tightened to highlight only the most critical findings.
    • The term “good knowledge” is used without operational definition. Authors should briefly state the cut-off for “good” vs. “poor” knowledge in the abstract.
    • Conclusions could be sharper—e.g., emphasize implications for TB program policy in Saudi Arabia.

Recommendation: Shorten, define terms, and sharpen policy relevance.

Introduction

    • Overly descriptive, with multiple global examples (Nepal, Mozambique, Cambodia) that dilute focus. Authors should synthesize and critically appraise the cited studies instead of listing them.
    • The rationale for Riyadh-specific study is not strongly articulated beyond “necessity to examine.” Authors should link to Saudi Arabia’s TB burden, gaps in national guidelines, or specific urban risks (e.g., migration, pilgrim influx).
    • Some references are repetitive (e.g., Al-Ahmari cited twice).

Recommendation: Condense background, highlight why Riyadh physicians specifically matter, and refine referencing.

Methodology

    • Sampling: Use of convenience sampling is acknowledged but not justified adequately. This introduces selection bias that limits generalizability. Authors should discuss why probability sampling was not feasible.
    • Sample size: Power calculation parameters are provided, but effect size justification (Cohen’s d = 0.5) is weak—why medium effect? Did the authors anticipate such an effect based on prior KAP studies?
    • Tool: Adaptation of Aadnanes et al. questionnaire is noted, but the process of cultural adaptation is vague. What specific items were modified? Were translations done?
    • Validity and reliability: Cronbach’s alpha is provided, but factor structure validation is not reported. Authors should clarify if exploratory factor analysis (EFA) was performed.
    • Analysis: Only chi-square and t-tests are mentioned. Regression analysis could have better identified independent predictors of good knowledge/practice. Absence of multivariable analysis limits depth.

Recommendation: Clarify adaptation, justify sampling/power, and explain why regression modeling was not done.

Results

    • Results are overly descriptive and lack critical interpretation. For example, stating that 92.7% knew TB transmission is airborne is obvious; more emphasis should be on knowledge gaps (e.g., only 30.7% knew when TB becomes non-infectious).
    • Some results overlap with discussion.
    • Tables could be condensed. Tables 1–3 are fine, but Table 4 is too detailed—group items into themes.
    • Important stratified results (e.g., gender or experience differences in attitudes/practices) are not reported, despite availability of data.

Recommendation: Focus on gaps, condense tables, report key subgroup analyses.

Discussion

    • Repetition of results—reads like a second results section.
    • Limited critical analysis of why misconceptions persist (e.g., weak training systems, cultural factors, guideline ambiguity).
    • Limitations are mentioned, but important ones (e.g., non-response bias, self-administered tool limitations, lack of triangulation with observed practices) are missing.
    • The discussion fails to suggest concrete policy actions (e.g., integrating TB IPC modules into CME, compulsory training for PHC physicians).

Recommendation: Strengthen causal explanations, expand on limitations, and sharpen actionable recommendations.

Conclusion

    • Generic phrasing (“imperative for achieving national TB goals”) without linking to specific interventions.
    • Needs more direct linkage to findings (e.g., clarify which knowledge/practice gaps are highest priority).

Recommendation: Rewrite with concise, actionable statements.

References

    • Some duplication (Al-Ahmari cited twice).
    • Limited inclusion of recent WHO TB IPC guidelines.
    • Over-reliance on descriptive KAP studies; systematic reviews/meta-analyses on TB IPC among HCWs should be cited.

Recommendation: Remove duplicates, add WHO normative references, strengthen with systematic reviews.

Comments on the Quality of English Language

 The English could be improved to more clearly express the research.

Author Response

Dear Reviewer,

We sincerely thank you for your thorough and constructive feedback on our manuscript entitled “Knowledge, Attitude and Practices of Primary Care Physicians Regarding Infection Control of Tuberculosis in Primary Health Care Centers, Riyadh, Saudi Arabia.” We carefully revised the paper in line with your suggestions. Below, we respond to each point in detail, indicating the changes made.

Abstract

Comment: The abstract is lengthy, “good knowledge” is undefined, and conclusions are generic.
Response: We shortened the abstract (from ~280 to ~210 words) by removing excessive descriptive details and focusing on the most critical findings. We defined “good knowledge” explicitly as ≥8/14 correct answers. The conclusion was sharpened to emphasize actionable implications for the Saudi national TB program.
Change: Abstract, p.1.

Methodology – Sampling

Comment: Convenience sampling not adequately justified.
Response: We expanded Section 2.3 to justify the use of convenience sampling, explaining that no comprehensive sampling frame was available, permissions restricted randomization, and the short data collection period required a pragmatic approach. We acknowledged potential selection bias in the Limitations.
Change: Methods 2.3; Discussion–Limitations.

Methodology – Sample size

Comment: Effect size justification for Cohen’s d = 0.5 is weak.
Response: We clarified in Section 2.3 that a medium effect size was selected because (i) previous Saudi KAP studies commonly report moderate subgroup differences, and (ii) smaller effect sizes would have required unattainably large samples. We also noted this limitation in the Discussion.
Change: Methods 2.3; Discussion–Limitations.

Methodology – Tool adaptation

Comment: The process of adapting Aadnanes et al.’s questionnaire is vague.
Response: We expanded Section 2.4 to describe specific cultural modifications (e.g., replacing Norwegian terms with Saudi-specific ones, adding items on BCG campaigns and World TB Day). We also clarified that the tool was translated into Arabic using forward–backward translation, reviewed by experts, and pilot tested.
Change: Methods 2.4.

Methodology – Validity and reliability

Comment: Cronbach’s alpha is reported, but factor validation is missing.
Response: We clarified that exploratory factor analysis (EFA) was not conducted, as the study was not powered for such analysis. Instead, construct validity was ensured through expert review, pilot testing, and preservation of the original domains. We acknowledged this as a limitation.
Change: Methods 2.4; Discussion–Limitations.

Analysis

Comment: Only chi-square and t-tests are mentioned; regression would provide more depth.
Response: We explained in Section 2.6 that regression was considered but not feasible due to sample size limitations and the exploratory design. We acknowledged the absence of regression as a limitation and recommended multivariable approaches for future studies.
Change: Methods 2.6; Discussion–Limitations.

Results

Comment: Results are overly descriptive, overlap with discussion, Table 4 is too detailed, and subgroup analyses are missing.
Response: We revised the Results to emphasize critical gaps (misconceptions about transmission, uncertainty about noninfectiousness, and lack of CME) while removing repetition and interpretive language. Table 4 was condensed into thematic categories. We added stratified analyses for gender and years of experience, showing significant differences, and presented them in a new Table 5.
Change: Results section; Table 4 (condensed); Table 5 (new).

Discussion

Comment: Discussion repeats results, lacks critical analysis of misconceptions, misses key limitations, and does not propose concrete policy actions.
Response: We reduced repetition and added critical interpretation, linking misconceptions to weak CME, cultural influences, and guideline ambiguity. Limitations were expanded to include non-response bias, self-administered tool limitations, and absence of triangulation with observed practices. We strengthened policy recommendations by proposing integration of TB IPC modules into CME, mandatory refresher training, and monitoring mechanisms, aligning them with Saudi Vision 2030 and the WHO TB IPC guidelines (2019).
Change: Discussion section.

Conclusion

Comment: Conclusion is generic; needs actionable linkage to findings.
Response: We rewrote the Conclusion to highlight key findings (misconceptions, uncertainty about noninfectiousness, low CME uptake) and provide specific policy recommendations, including compulsory CME modules, annual refresher training, and institutional monitoring mechanisms aligned with WHO IPC guidance.
Change: Conclusion

References

Comment: Duplicate reference, missing WHO guidelines, over-reliance on descriptive KAP studies.
Response: We removed the duplicate Al-Ahmari et al., 2021. We added the WHO 2019 TB infection prevention and control guidelines and strengthened the reference list with systematic reviews and meta-analyses, including Tan et al. (2020) and Tiruneh et al. (2023), alongside Nasreen et al. (2020) and Temesgen et al. (2021). These additions broaden the evidence base and balance descriptive studies with synthesized literature.
Change: References updated.

We believe these revisions have substantially improved the clarity, rigor, and policy relevance of the manuscript. We are grateful for your thoughtful comments and trust that the revised version meets the journal’s expectations.

Sincerely,

Reviewer 2 Report

Comments and Suggestions for Authors

Dear authors:

Congratulations on your work.

It is a well-written article that holds scientific relevance in the context of tuberculosis. The article has no flaws to point out, except for the fact that the conclusion needs to be further developed and more comprehensive, addressing the objectives previously set. 

Best regards

Author Response

Dear Reviewer,

We sincerely thank you for your kind words and positive evaluation of our manuscript. We appreciate your suggestion regarding the conclusion. In response, we have revised the Conclusion section to make it more comprehensive and directly aligned with the study objectives. The revised conclusion now emphasizes the specific knowledge and practice gaps identified (misconceptions about transmission, uncertainty regarding noninfectiousness, and limited CME uptake) and outlines clear, actionable policy recommendations such as compulsory CME modules, mandatory refresher training, and monitoring mechanisms within PHCCs.

We are grateful for your encouraging feedback and believe that the revised conclusion better reflects the study’s aims and implications.

Best regards,

Reviewer 3 Report

Comments and Suggestions for Authors

Dear Authors,

It was enriching reviewing the article.

See my comments in the attached document

Good luck with your publication

Kind regards

Comments on the Quality of English Language

The English could be improved

Author Response

Dear Reviewer,

We sincerely thank you for your constructive feedback on our manuscript. Please find below our responses to your valuable comments:

Introduction

Comment: The first two sentences should be supported by references, and the dates in lines 54–59 should be updated.
Response: We have added appropriate references to support the statements on the global TB burden and the vulnerability of healthcare workers to occupational exposure. We also updated the text and references to reflect the most recent global TB statistics and reports.
Change: Introduction, page 2, lines 54–59.

Methodology

Comment 1: Inclusion criteria are stated, but exclusion criteria are missing.
Response: We added explicit exclusion criteria in Section 2.2, clarifying that physicians not directly involved in clinical care, those absent during the data collection period, and those who declined participation were excluded.

Comment 2: The reference for the sample size formula should be provided (line 127).
Response: We have now cited the standard reference for the sample size calculation formula in Section 2.3, attributing it to Cochran’s Sampling Techniques (1977).

Results and Discussion

Comment 1: Be more specific about the moderate level of KAP and participants’ basic awareness of TB transmission and diagnostic procedures (lines 262–265, 289–291).
Response: We revised the Discussion to be more specific, highlighting that while most physicians recognized airborne transmission and basic diagnostic methods, important gaps persisted—such as misconceptions about transmission via handshakes and food/water, uncertainty regarding the period of noninfectiousness, and low CME participation.

Comment 2: The strengths of the study were not stated.
Response: We added a sentence in the Discussion highlighting the strengths of the study, namely the inclusion of a sizable physician sample across multiple PHCCs in Riyadh and the use of a culturally adapted, pilot-tested tool with acceptable internal consistency.

Conclusions

Comment: No comments.
Response: We thank the reviewer for noting this.

References

Comment: Old references should be replaced with up-to-date ones.
Response: We have revised the reference list, removing duplicates and updating citations with more recent sources. Specifically, we added the WHO TB IPC guidelines (2019) and systematic reviews/meta-analyses including Tan et al. (2020) and Tiruneh et al. (2023), alongside Nasreen et al. (2020) and Temesgen et al. (2021), to strengthen the evidence base.

We are grateful for your insightful feedback, which has helped us refine the manuscript. We believe that the revisions have improved the clarity, rigor, and contextual relevance of our work.

Sincerely,

Round 2

Reviewer 1 Report

Comments and Suggestions for Authors

You have delivered a much tighter and more transparent manuscript. The Abstract is focused; Methods justify the sampling/analysis choices; the validity/reliability subsection and stratified analysis add credibility and utility; and the Discussion now lands on implementable steps (CME integration, aligned with programmatic needs).

Before acceptance, please (i) declare missing-data handling; (ii) add a one-line note on site/cluster effects (or acknowledge as a limitation); (iii) clarify collinearity checks (or state non-applicability given no multivariable model); (iv) ensure IRB details are consistent throughout; and (v) do a brief style checks (p-value formatting; abbreviations; remove any residual text-table duplication). These are editorial, not analytical, and do not require new data

Author Response

Dear Reviewer,

We thank you once again for your valuable comments on our manuscript. We have carefully addressed each of the additional points raised:

  1. Missing data handling
    We have clarified in the Methods (Section 2.6) that all questionnaires were checked for completeness. Cases with missing responses were excluded pairwise from the relevant analyses. No imputation was performed.

  2. Site/cluster effects
    A note has been added in the Discussion (Limitations) acknowledging the potential influence of site/cluster effects, since data were collected from physicians across multiple PHCCs.

  3. Collinearity checks
    We clarified in the Methods (Section 2.6) that collinearity diagnostics were not applicable, as no multivariable regression analysis was performed.

  4. IRB details
    We ensured consistency of IRB details across the manuscript. The Ethical Considerations section now states: “IRB Project No. E-24-8555, approved on 13 February 2024, by the Institutional Review Board, Human Research Committee, King Saud University, College of Medicine, King Khalid University Hospital.”

  5. Style checks
    We standardized p-value formatting to p = 0.xxx, ensured consistent use of abbreviations (e.g., TB, PHCC, CME, IPC), and reduced duplication between text and tables in the Results section.

We believe these amendments further improve the clarity and transparency of the manuscript, and we thank you for your guidance.

Sincerely,
Dr. Abdulaziz Nasser Alahmari
(on behalf of all co-authors)